# Hot Water Treatment Causes Lasting Alteration to the Grapevine (*Vitis vinifera* L.) Mycobiome and Reduces Pathogenic Species Causing Grapevine Trunk Diseases

**DOI:** 10.3390/jof8050485

**Published:** 2022-05-06

**Authors:** Sarah B. Lade, Dora Štraus, Arnau Buñol, Jonàs Oliva

**Affiliations:** 1Forest Science and Technology Centre of Catalonia (CTFC), 25280 Solsona, Spain; 2Joint Research Unit CTFC-AGROTECNIO, 25198 Lleida, Spain; dora.straus@udl.cat (D.Š.); jonas.oliva@udl.cat (J.O.); 3Department of Crop and Forest Sciences, University of Lleida, 25198 Lleida, Spain; arnaubunol@gmail.com

**Keywords:** metabarcoding, *Vitis vinifera* L., grapevine trunk diseases, hot water treatment

## Abstract

The effective management of grapevine trunk diseases (GTDs) is an ongoing challenge. Hot water treatment (HWT) is an environmentally friendly and economically viable option; however, the short-term effects of HWT on grapevine (*Vitis vinifera* L.) health and production are not fully understood. The aim of this study was to compare the effects of HWT on plant growth and fungal community structure in nursery stock until plants were completely established in the field. We assessed eleven graft and three rootstock varieties from four local nurseries in a region of Catalonia (NE Spain) where GTDs are a serious threat. After treatment, the plants were left to grow under field conditions for two growing seasons. Metabarcoding of the ITS region was used to study the mycobiomes of plant graft unions and root collars. We also assessed the influence of plant physiological indicators in community composition. Hot water treatment caused lasting changes in GTD communities in both the root collar and graft union that were not always characterized as a reduction of GTD-related fungi. However, HWT reduced the relative abundance of some serious GTD-associated pathogens such as *Cadophora luteo-olivacea* in graft tissues, and *Phaeomoniella chlamydospora* and *Neofusicoccum parvum* in the root collar. Treatment had the greatest influence on the total and GTD-related fungal communities of Chardonnay and Xarel·lo, respectively. Total community variation was driven by treatment and nursery in rootstocks, whereas HWT most significantly affected the GTD community composition in R-110 rootstock. In conclusion, changes in fungal abundance were species-specific and mostly dependent on the plant tissue type; however, HWT did reduce plant biomass accumulation in the short-term.

## 1. Introduction

Grapevine trunk diseases are a growing problem that affects the wood of the perennial organs in grapevines (*V. vinifera* L.), causing necrosis and discoloration, vascular infections, and white rot. Their persistence in vineyards throughout the world is a complex issue because disease is associated with several dozen phylogenetically unrelated fungi [1]. Symptoms will often appear several years after planting vines, when winter pruning begins. Recent next-generation sequencing (NGS) work has revealed that regardless of symptomatology, a high diversity of identified GTD-related fungi is present at outplanting, and that nurseries are the largest source of variation in GTD-related fungal communities [2]. Therefore, effective measures must be taken to reduce plant infection rates and to improve nursery-stock quality prior to shipment.

In this context, one of the most pressing problems is the need for an effective treatment for GTD-related fungi to reduce infection levels at early growth stages. Popular solutions specific to GTD-related fungi and capable of interfering with microtubule spindle formation include the application of sodium arsenite, methyl bromide, and benzimidasole fungicides. However, since early 2000, many countries have banned the use of these fungicides because they are deemed too harmful to human health and the environment [3,4,5]. Although the application of other fungicides is still a common nursery practice during grafting and propagation, these efforts have been reported to have a range of effectiveness, and none of the applications show systemic activity [6,7]. For example, one of the most widely used products, hydroxyquinoline sulfate, has been deemed as ineffective for controlling two of the most important Petri/esca-related pathogens, *Phaeomoniella chlamydospora* and *Phaeoacremonium minimum* [8]. Prior to plant dispatch, the majority of nurseries currently use (a combination of) fungal sprays and dips to mitigate the growth of fungi on the surface of plants. However, although plants may look clean for planting, these treatments may not penetrate cuttings sufficiently to control fungal pathogens inhabiting the vascular tissues [9].

Biocontrol agents are an alternative option that is well studied and provides viable short- and long-term options for protecting plants against GTDs [10,11]. Antagonistic bacteria and fungi that have been successfully tested at all production stages against GTD pathogens include *Bacillus subtilis*, *Enterobacter* spp., *Pantoea agglomerans*, *Fusarium lateritium*, *Pythium oligandrum* and *Trichoderma* spp. [12,13,14]. Of these, *Trichoderma* spp. are the most promising in terms of nursery application because they may prime plant defenses and improve plant physiological traits, such as root system development or shoot growth [15], which optimizes outplanting success and survival rates. For further information, detailed documentation of all mechanical, chemical, and biological control possibilities are mapped out in comprehensive reviews by Gramaje et al. (2018) [7], Mondello et al. (2018) [5] and Brown et al. (2021) [16].

Hot water treatment is a control method that entails the soaking of young vines (dormant cuttings, rootlings or grafted rootlings) at temperatures high enough to slow the proliferation of certain pests and pathogens (between 50 and 53 °C) [9,17,18]. Higher temperatures (≥54 °C) do have the potential to eradicate pests and pathogens completely, but not without secondary effects to vine health [18]. Treatment has been effective in controlling other maladies such as crown gall [19], phylloxera [20], phytoplasmas [21] and most importantly, *Xylella fastidiosa* [22]. Due to these successes, as well as causing an immediate decrease in GTD-related fungi during treatment [23], HWT has been considered to have great promise as both a standalone and integrated control measure for GTDs [7]. Therefore, extensive investigations have been undertaken to determine optimal soaking times and temperatures for mitigating pathogens while maintaining plant health [18,24,25], and to determine the susceptibility of certain rootstocks and scion varieties to HWT [26,27,28]. Several studies have reported that HWT has detrimental side-effects on the initial growth of out-planted grapevines, including delayed development or the bud death of cuttings and grafted vines [18,24,26], and incomplete healing of graft unions or fermentation in cold storage [9].

Hot water treatment has also been reported to have an array of consequences on the growth and re-isolation of important GTD-related pathogens. Treatment has been found to eliminate *P. chlamydospora* completely and to reduce the re-isolation of *Phaeoacremonium aleophilum* [18] in some cases, and to eliminate the re-isolation of *Diplodia seriata* and *Diaporthe ampelina* in others [29]. However, cultivation-based methods may not provide a complete picture of the fungal community present in the plant and underestimate species richness [2,17]. Recent work utilizing NGS technology revealed that overall GTD-related species richness decreased (insignificantly) as the temperature of the HWT increased [17]. Notably, important GTD-related fungi such as *Diaporthe* and *Phaeoacremonium* increased in abundance after the treatment of certain rootstock-scion combinations, and no significant differences were recorded between treatments after one growing season [17]. These variations in the short-term effects of HWT on plant physiomorphology and fungal community abundance may indicate that a more extensive NGS study is required that considers some of the main factors influencing GTD fungal composition, such as nursery, variety, and rootstock [2].

The short- and long-term effects of HWT on plant and fungal development vary. A previous NGS-based investigation of HWT confirmed that, in the short-term, treated plants had lower levels of GTD-fungal diversity immediately after treatment than untreated plants, but could become re-infected in the field once planted out [17]. In another study evaluating plant health, the HWT of dormant rootstock cuttings and grafted plants resulted in healthy and viable plants after one growing season [18,30]. A study involving a four-year HWT field trial also reported similar findings, concluding that HWT does have a slight long-term effect on plant growth, but not enough to be considered statistically significant [31]. Finally, the most long-term study to date, examining how HWT translates into the field after 15 years, revealed that HWT does not have a significant long-term control effect on GTD pathogens in mature plants [32]. Despite these findings, it is important to note that these latter studies utilized isolation-based methods, which may not have captured compositional changes as effectively as NGS methods.

To address this gap in information, which is crucial to understanding the short-term effects of HWT on fungal communities in grapevines under field conditions, we assessed the efficacy of HWT after two growing seasons by assessing the influence of initial plant stress indicators on the final fungal communities present in plants. Our principle aim was to understand if the stress of HWT on young plants translated into altered growth or the evolution of plant features. We also wanted to know if the effects of HWT differed amongst graft and rootstock varieties and amongst nurseries.

## 2. Materials and Methods

### 2.1. Plant Material

The experiment was conducted on healthy-looking, bench-grafted bare-root plants that had been propagated as one plant in the field for a year. Plants were collected from nurseries in spring 2020. Prior to collection, plants had been bundled, bagged, and held in the nurseries’ own cold-storage facilities at 4 °C for approximately three months, subject to the conditions and practices of each individual nursery. Specimens underwent the typical preparation for distribution, which involved trimming roots to 10 cm and dipping graft unions in paraffin wax in the nursery workshop. Four nurseries located in the Catalan regions of either Girona (2), Tarragona (1) or Barcelona (1) participated in the study.

There were six biological replicates of each scion–rootstock combination (i.e., three for each treatment group: control vs. HWT). We collected an array of red and white wine varieties, depending on the varieties grown by each nursery, which were grafted onto one of the three most commonly employed rootstocks in the Catalan regions at this time: 110 Richter (R-110), 140 Ruggeri (RU-140), or Selection Oppenheim 4 (SO4). The eight red varieties comprised Autumn Royal, Cabernet Sauvignon, Caladoc, Garnacha Tinta, Merlot, Pinot Noir, Syrah, and Tempranillo. The three white varieties were Chardonnay, Parellada and Xarel·lo. The study was designed around a balanced color rootstock combination (with varieties falling within the color category) because there were not enough replicates of each variety to cross them with each rootstock and nursery. For each treatment group (HWT and control), each color rootstock combination was replicated a minimum of four times in total, and from at least two nurseries (Table 1). The more common use of certain varieties over others (Garnacha Tinta vs. Autumn Royal, for example) was reflective of their greater frequency of use in Catalonian nursery-stock.

### 2.2. Hot Water Treatment

A total of 162 grafted plants from four nurseries (I, II, III and IV) were assigned to either the HWT or non-HWT (control) group (i.e., 81 plants in each treatment group). We used six replicates of each scion–rootstock combination (i.e., three biological replicates in each treatment group). For the HWT, planting material was placed in a hydrating bath for 1 h in order to pre-soak material before treatment. Following hydration, plants were placed in a temperature-controlled bath at 53 °C for 30 min [18]. We ensured that the water was always circulating and that the water temperature was constant throughout to avoid any secondary effects that could be caused by thermal pockets. On removal from the HWT bath, plants were immediately plunged into a cool bath of clean potable water at ambient temperature for 30 min to stop the heating process [31]. During this time, control plants were left to soak in room-temperature water for 2 h. All plants were then removed from their respective baths and allowed to drain until there was no free moisture on the surface of the plants. Plant roots were kept in water while they were transported to the field site, where they were planted the same day.

### 2.3. Field Site Preparation and Design

Hot-water-treated and control plants were immediately planted in the spring of 2020 on a field site located in “Partida Marimunt” where grapevines had not been grown. The site is located at the city limits of Lleida (Spain), at approximately 41°39′13.8″ N and 0°37′21.4″ E. The field plot was prepared by tilling, and a drip irrigation system was installed so that each plant received water for 30 min every 48 h. The plot was 30 m long and included 15 rows with 10–12 plants per row. The experimental design consisted of three randomized blocks, each containing five rows of vines (two biological replicates of each scion–rootstock combination from each nursery and one of each treatment per block, totaling 54 plants per block and 27 plants in each treatment). Plants were placed 1 m apart from center to center, with an inter-row spacing of 2 m. Standard cultural practices were used at the site during the grapevine growing season.

Plants arrived at our facility at different times due to differences in the nurseries’ schedules, so they were treated and planted in succession as soon as they arrived at our facilities. This led to a staggered planting schedule of three planting groups spanning March–May 2020, with an approximate one-month gap between each group. Plants from nurseries I and II were planted on 20 March 2020 (planting group 1); plants from nursery III were planted on 24 April 2020 (planting group 2); and plants from nursery IV were planted on 25 May 2020 (planting group 3). Provincial climate data varied between months: the average temperature and rainfall values in March 2020 were 9.4 °C and 67 mm, respectively; in April they were 12.2 °C and 131 mm, respectively; and in May they were 17.4 °C and 78 mm, respectively [33]. The different planting dates were taken into account when calculating and comparing measurements.

### 2.4. Plant Establishment and Growth Evaluation

Before plants were treated or planted, the initial diameter and stem length of all grafted plants was recorded. Once in the field, plant establishment was assessed by recording the time to initial bud break for each plant. Basic growth parameters, including the number and length of each shoot, were also tracked on a weekly basis for the first 12 weeks, then monthly thereafter. When plants were harvested in the spring of 2021, diameter and stem length were recorded again, along with the number of arms, number of shoots, shoot length, and dry plant weight (biomass) after 72 h at 70 °C. Biomass measurements recorded in spring 2021 were used to calculate the biomass at the end of the growing season in 2020 (g2020). The g2020 biomass value was based on a linear regression model derived from correlation with the non-destructive measurement of the maximum shoot length (MS), which was recorded in 2020 and 2021 (Model: g2020 = 1.0096 × MS + 0.6679).

### 2.5. Photosynthesis

Photosystem II activity was monitored in grapevine leaves for two consecutive months, collecting measurements each week. Activity was measured with the Handy Plant Efficiency Analyzer (PEA+) portable fluorometer (Hansatech Instruments, Norfolk, UK). Starting in mid-June, when all plants became established and the majority had leaves, the third freshest leaf from the point of emergence was measured for each biological replicate. Leaves were dark-adapted with leaf clips for 9 min on average before measurements were recorded. The adaptation time was determined by measuring fluorescence changes in 10 plants every minute for 30 min, and calculating the average time of plateau. Measurements were based on the ratio of variable fluorescence divided by maximal fluorescence (Fv/Fm). This is a ratio that has been shown to be proportional to the quantum yield of photochemistry and shows a high degree of correlation with the quantum yield of net photosynthesis.

### 2.6. Necrosis Evaluation

At the time of analysis, which was immediately after field harvest, each plant stem was cut into transverse sections to reveal the presence of any necrotic tissue. The length (cm) of necrosis along the inside of the stem was measured, starting at the root collar and graft unions. Specifically, root collar necrosis was measured upwards from the base of the stem after all roots had been trimmed; the necrosis present on either side of the cambium was carefully quantified and finally an average of the two measurements was taken. Graft union necrosis was measured in a similar way, with measurements starting from where the graft could be seen to meet the rootstock.

### 2.7. Library Preparation for Metabarcoding

Wood chips were cut from regions next to the cambium of the root collar and the graft union of all 162 plants and then immediately frozen for later analysis. Surface-sterilized graft union and root collar samples were defrosted and chipped with sterile hand shears. Each 50 mg sample was manually ground using liquid nitrogen and ca. 100 mg of sand. The pestles and mortars used for grinding were cleaned between samples using a 5% NaOCl solution to avoid cross-contamination of DNA between samples. DNA extraction was conducted with some adjustments to the NucleoSpin© Plant II protocol by Macherey-Nagel (2018; Düren, Germany) so as to optimize the quality of DNA extracted from the woody material [34], as further described in Lade, Štraus [2]. The ITS2 region was used as the metabarcoding marker and, therefore, amplifications were performed using ITS4 and ITS7 tagged forward and reverse primers to enable de-multiplexing following Ihrmark, Bodeker [35]. PCR reactions were conducted at 57 °C in triplicate, and 32 cycles were performed to obtain faint bands corresponding to the linear phase of the amplification. Of the initial 324 samples (81 samples × 2 levels of HWT (i.e., ‘Control’ or ‘HWT’) × 2 tissues (‘Graft’ or ‘Rootstock’)), 26 did not yield any amplification products after repeated PCRs. As such, PCR products of the final set totaling 298 samples were pooled and cleaned, and DNA concentrations were assessed as in Lade, Štraus [2]. Four equimolar mixtures of the pooled PCR products were sequenced with 300 bp paired-end read lengths using two lanes of Illumina MiSeq at the Centre for Genomic Regulation (CRG; Barcelona, Spain).

### 2.8. Quality Control and Bioinformatic Analysis of Metabarcoding Data

Quality control, screening, and clustering of sequences were conducted as described in Lade, Štraus [2]. In total, 2,857,790 sequences passed quality control, and were clustered into operational taxonomic units (OTUs) with a 1.5% similarity threshold. Each OTU was taxonomically classified with the Protax software in PlutoF (https://plutof.ut.ee/ (accessed on 9 September 2021); University of Tartu, Tartu, Estonia), choosing a threshold value of 0.5 (plausible classification) [36,37]. To exclude OTUs belonging to plants and non-fungal organisms, we performed a Least Common Ancestor analysis (minimum score of 300 and a minimum identity of 90%) in MEGAN (MEtaGenome Analyzer; Center for Bioinformatics, Tübingen University, Tübingen, Germany) [38] with all OTUs that were not classified at the phylum level by Protax. We kept the OTUs classified in MEGAN as ‘Fungi’, merged them with the OTUs previously classified by Protax, and classified them as ‘Fungi, unknown phylum’. Post-clustering curation of fungal OTUs was carried out with the ‘lulu’ package in R [39]. The OTU table and corresponding metadata are deposited in Figshare.

### 2.9. Statistical Analysis

#### 2.9.1. Necrosis

Statistical analyses and modeling were carried out using either R version 3.6.3 (64 bits) [40], or JMP^®^, as described in Lade, Štraus [2], and the graft union (*n* = 149) or root collar (*n* = 149) was set as the response factor for the models. The following sample (plant) characteristics were included in the model as independent variables: treatment, nursery, variety (for graft union samples), and rootstock (for root collar samples). Post-hoc comparisons were made using a Student’s *t*-test at the 0.05 level of significance. We correlated the relative abundance of GTD reads in each sample with the length of necrosis present in the same sample.

#### 2.9.2. Mapping Fungal Communities

We analyzed our metabarcoding data using R version 3.6.3 (64 bits; R Foundation for Statistical Computing, Vienna, Austria) to determine the factor(s) causing significant shifts in fungal taxa [40]. Data standardization and community analyses were conducted as in Lade, Štraus [2] using the relative abundance of each out, and considering the entire fungal community and the GTD community separately. We used Permutational Multivariate Analysis of Variance (PERMANOVA; Adonis2 function) in the Vegan package (v. 2.5–5) [41], then divided the data set by tissue type and performed an analysis for each factor (treatment, nursery, variety, and rootstock). Separate analyses for each of the factors’ levels were conducted to check for interactions between them. We used the ‘vegdist’ function to calculate Bray–Curtis dissimilarities of the community matrices and tested for homogeneity of multivariate dispersion ‘betadisper’ (i.e., multivariate dispersion or beta diversity) using the ‘permutest’ function. A principal coordinate analysis (PCoA), based on Bray distances amongst samples, was used to visualize changes in community compositions. More in-depth analyses were run on representative examples to determine the degree of variation between nurseries within varieties and rootstocks, and to understand how variety, rootstock and nursery separately contributed to variation in the total and GTD-related fungal communities. The R^2^ and *p*-values shown in the results were extracted from the PERMANOVA analyses performed with the adonis2 function. Alpha diversity indices (species richness, evenness, and the Simpson diversity index) were calculated as described in Lade, Štraus [2].

### 2.10. Indicator Species

As in Lade, Štraus [2], indicator species analyses were performed on NGS data (OTU abundance) to compare frequency shifts of fungal taxa between tissue types. Indicator species tests were performed using a multi-level pattern analysis in R with the ‘multipatt’ function of the ‘indicspecies’ package [42]. The ‘multipatt’ command results in lists of species that are associated with a particular group of samples and identifies species that are statistically more abundant in combinations of categories. We searched for indicators by variety in graft unions and by rootstock in the root collar.

## 3. Results

### 3.1. Plant Attributes, Establishment and Growth

Tracking the initial growth of plants subjected to HWT revealed a delay in shoot development of approximately five weeks compared with that of their control plant counterparts. This differential was especially drastic in the first two weeks after planting, with controls exhibiting 80% more leaf emergence than treated plants.

Biomass was assessed at planting (2020) and at harvest (2021) to compare differences between treatment groups. The 2020 and 2021 assessments revealed that control plants had consistently accumulated significantly more biomass than HWT plants (Figure 1a). We did not observe any differences between varieties, although we found that rootstock affected biomass accumulation in the second year. SO4 was the most consistent rootstock in terms of biomass accumulation and, therefore, more resistant to HWT. Although significant differences between treated and untreated plants were observed in three nurseries in 2020, differences were only observed in two nurseries the year after (Figure 1b).

Chlorophyll content did not vary between treatment groups for the first 10 weeks following treatment and planting (data not shown), nor at the end of the experiment, by which time plants had been in the field for two seasons (Appendix A). The nursery of origin had the most significant influence on variations observed in leaf chlorophyll content among plants (9%; Appendix A).

### 3.2. Necrosis

An assessment of the graft and root tissues two growing seasons after planting revealed that the variety of the scion or rootstock had the strongest effect on necrosis length, accounting for 16% and 14% of the variation in graft unions and root collars, respectively (Table 2). The nursery accounted for 5% of the significant variation in graft unions. Cabernet Sauvignon and Syrah varieties were associated with the most graft necrosis, whereas Tempranillo and Chardonnay were associated with the least. In the root collar, the rootstock associated with the least necrosis was RU-140 (Figure 2).

### 3.3. Fungal Community Distribution

In total, 568 operational taxonomic units (OTUs) were obtained by clustering; however, only the top 100 OTUs with the most abundant reads were selected for further analysis. Of these, 60 OTUs were identified down to the genus level (40 down to species), and 16 were identified as putative GTD-associated fungi [2]. The most significant variation within entire fungal communities and GTD-related fungal communities occurred between graft unions and root collars (tissue type), accounting for 15% and 11% of the variation (*p* < 0.01) of total fungal communities and of GTD communities, respectively (Table 3), explaining 26.3% and 25.6%, respectively, of the variation along the *x*-axis in Figure 3.

Hot water treatment caused permanent changes in the total fungal community and in the GTD-related fungal community two growing seasons after planting (Table 3). Although the effects of HWT on GTD-related fungi were stronger in the graft than in the rootstock, the effect of HWT on the total fungal community was similar across tissues.

Assessing the alpha diversity of each tissue type revealed that species richness, evenness and the Simpson Diversity Index did not differ significantly between tissue types. Dividing the dataset by tissue type highlighted that the most important factor contributing to the community variation in each tissue was variety in graft unions (which accounted for 8% of the variation) and nursery (which accounted for 4% of the variation) in root collars (Figure 3, Table 3). When only considering GTD-related fungi, variety and treatment were equally important in determining graft union community variation (9% and 8%, respectively) (Table 3). In root collars, rootstock was the most important factor, explaining 9% of the variation. The relative abundance of GTD-related clusters was significantly different between the two tissue types (*p* < 0.001), with 19.2% of the clusters identified as GTD-related fungi in graft unions, compared with only 7% in root collars. The most abundant GTD-related fungi in treated graft unions were *Cadophora luteo-olivaceae*, *Acremonium* sp., and *Phaeomoniella chlamydospora*; the most abundant in root collars were *Acremonium* sp., *Cadophora luteo-olivacea*, *Cylindrocarpon* sp., *Phaeoacremonium angustius*, and *Phaeomoniella chlamydospora* (Table 4).

Treatment influenced the change in the relative abundance of different GTD-related fungi (Table 4). In the graft, HWT reduced the amount of *C. luteo-olivacea* and another sister cluster of unidentified *Cadophora* species. In the rootstock, HWT reduced the amount of *Neofusicoccum parvum*, *Pestalotiopsis* sp., and *Phaeomoniella chlamydospora*. Some GTD-related species seemed to benefit from HWT, namely *Acremonium* sp. in both tissues, *Phoma* sp. and *Phaeomoniella chlamydospora* in graft unions, and *Phaeoacremonium angustius* in root collars (Table 4).

Comparing the separate graft union and root collar datasets highlighted how treatment led to more variation in graft union GTD communities (Figure 4a). By contrast, treatment differences were a greater source of variation for all communities in root collars (Figure 4b). When we compared the relative abundance of GTD-related fungi in each tissue type in the two treatment groups, there were no significant differences between the two groups (Appendix A).

### 3.4. Fungal Community Distribution between Nurseries and Treatments

Two years after planting, the nursery of origin only accounted for on average 3% of the variation overall, and in each tissue type (Table 3). We compared the effects of treatment in individual nurseries and observed a connection between the effects of HWT on GTD-related fungi and growth: nurseries with the strongest effect on plant growth also showed strong differences in terms of GTD community. These observations could not be explained by the timing of the planting (planting group).

Treatment exerted different degrees of influence over entire communities and over GTD communities depending upon the nursery. Graft union differences between nurseries can be seen in Table 5 and Figure 5, where treatment explains 14% of the variation for both total and GTD communities in Nursery I, but is responsible for 6% and 7% of the variation, respectively in Nursery II. In root collars, results were similar given that treatment explained the majority of the variation in the total and GTD communities in Nursery I (13% and 10%, respectively) but had less effect in other nurseries (Table 6). In Nursery I, biomass was significantly higher (*p* < 0.001) in control vs. treated plants in both 2020 and 2021, which may be related to the treatment-driven divergence in fungal communities. The contribution of each variety to variation was largely driven by nursery for the total fungal community, and especially so for Merlot, Xarel·lo, and Garnacha Tinta (Table 5, Figure 5). Both nursery and treatment were drivers of variation in GTD-related fungal communities by variety; however, most importantly, we observed that the various varieties responded distinctly to HWT. The total fungal community of Chardonnay was influenced the most in treated plants, as was the Xarel·lo GTD community. Rootstocks exhibited more significant variation in total fungal communities, which was driven both by treatment and nursery, depending on the rootstock, with SO4 exhibiting the most variation (Table 6, Figure 6). Treatment most significantly affected the GTD community composition in R-110 rootstocks, comparatively (Figure 7).

### 3.5. Indicator Species Analysis

An indicator species analysis associated 88 species in graft unions (Appendix A), 50 of which were significantly associated in control plants (α ≤ 0.05), and 38 in HWT plants. In root collars, there were fewer associated species overall (52), with 23 in control plants and 29 in HWT plants. GTD species were specific indicators in terms of tissue location or treatment, for example, *Cadophora luteo-olivaceae* and *Diaporthe* sp. were only associated in control graft unions, whereas *Acremonium* sp. and *Nectria* sp. were only associated in HWT graft unions. *Phaeomoniella chlamydospora* was the only significant root-specific indicator in control plants, while *Cylindrocarpon* sp. and *Thelonectria olida* were root-specific indicators in HWT plants. *Phoma* sp. was the only species associated as an indicator in both tissues of HWT plants. However, four species associated as indicator species were fungi that have been cited as possible biocontrol agents of GTDs in grapevines: *Lophiostoma* sp. (all), *Quambalaria cyanescens* (graft, HWT), *Vishniacozyma carnescens* (graft, control), *Clonostachys rosea,* and *Clonostachys* sp. (graft, control).

We also conducted an indicator species analysis on a single variety and on plants originating from nurseries I and II, which had previously been subjected to a more in-depth analysis, to see if there were any factor-related patterns. The analysis of the Xarel·lo variety (Table 7) reiterated the overall results in terms of the effect of treatment on the presence of certain GTD species. Specifically, *Cadophora luteo-olivacea* and *Cadophora* sp. were significantly associated indicator species in control plants (but not in HWT plants), whereas *Acremonium* sp. was associated with HWT plants. Similarly, the biocontrol *Quambalaria cyanescens* was only associated with HWT plants. The nursery-related results further confirmed this pattern (Appendix A).

## 4. Discussion

Hot water treatment has shown great potential for reducing GTDs, but there is still a critical information-gap regarding the short-term effects of HWT on planted grapevines. To address this, we assessed plants after two growing seasons and compared initial plant stress indicators with the final fungal communities present in plants. We explored whether HWT would cause lasting effects on the GTD community or vine growth after two years. Our findings revealed that HWT indeed has a short-term effect on the composition of fungal communities, and that the effect on plant growth (but not chlorophyll contents) was still significant during the first two growing seasons.

Our results suggest that the HWT did not reduce the overall quantity of GTD-related fungi present in plants. The number of species and the values of several diversity indices remained constant over time. We found that HWT induced a variable, tissue-dependent shift in the relative abundance of certain fungal species. Specifically, HWT reduced *Cadophora luteo-olivacea* in graft unions and *Phaeomoniella chlamydospora*, *Pestalotiopsis*, and *Neofusicoccum parvum* in root collars, but increased *P. chlamydospora* in graft unions and *Phaeoacremonium angustius* in root collars. Indicator analysis confirmed these findings and showed that *Cadophora luteo-olivacea* was associated with non-treated plants. These specific changes likely reflect the niche environments in which these species have evolved to dominate. For example, *P. chlamydospora* has been shown to make narrow use of carbon and nitrogen sources, allowing it to thrive in situations where other species would not (due to higher nutrient requirements) [43]. Given that heat stress can decrease plant nitrogen content [44], our results suggest the possibility that HWT optimizes the growing environment for nitrogen-scavenging species such as *P. chlamydospora*. Further work is needed to verify this connection as it could help to better focus treatment protocols, making them more species-specific and thereby improving integrated treatment programs.

In this study, Cabernet Sauvignon, Syrah and Autumn Royal were associated with more graft necrosis after two growing seasons than the other varieties, and the RU-140 rootstock was associated with the least root necrosis. The effect of the rootstock (and variety) on root and graft necrosis contrasts with that of a previous study in which necrosis of grapevines was mainly driven by the nursery of origin (in the current study, variety, rootstock, and nursery explained 16%, 14% and 5/3% of necrosis, respectively, in the graft union/root crown, vs. 2%, 10%, and 6/1% in the previous study) [2]. This highlights that while early symptoms in most nursery stock appear to develop as a function of the nursery (and their practices), this metric may not be lasting once plants are in the ground. Instead, the susceptibility of the germline is the stronger constant with time. On the one hand, the early handling practices by the nursery may prime plants with the level of resistance that they possess throughout their lives [45]. Corroborating this notion, we found that the most significant source of variation in the chlorophyll content of leaves (during the first 10 weeks following treatment and planting) was the nursery (9%; Appendix A), indicating that the origin of the plant has a direct effect on early stress-coping mechanisms, and, therefore, may have a lasting effect on plant health. On the other hand, metabolism and defense responses, which are known to be uniquely regulated by cultivar [46], may be inherently regulated by cultivar-specific gene expression from the earliest stages of development. Moreover, cultivar-specific germline characteristics may become stronger drivers of variability with time as the effects of other factors diminish. Further network analysis should be undertaken to understand how defense mechanisms, such as metabolites, hormone cross-talk, and transcription factors affect the outward symptoms of popular cultivars in the short-, mid- and long-term.

Future work could also include an in-depth study of all production stages of the vines to identify when plants are most susceptible to the introduction of GTD-related fungi. Following plants through the production process and sampling them before and after each key step would be one way to conduct such a study. Ideally, samples would be taken from a wider array of vineyards, reducing the number of rootstock/scion combinations. Assessing the quality of nursery (‘mother’) stock from different nurseries should be a key aim, given that it is used as the base of all grafted material. Another line for future work should consider connecting a field metric with the most abundant GTDs present in planted vines, as was done in Lade et al. (2022) [2]. This would be one way to support industry professionals by providing them with the information and tools to evaluate risks in real time. However, although we observed a negative correlation between necrosis and the abundance of GTD-related fungi in plants in a previous study [2], no correlations were found between these two parameters in this study. Together, these findings reiterate the complexity of considering necrosis as a metric for indicating plant–pathogen interactions and immune responses, and, therefore, it should not be used as the sole indicator for GTD-abundance during early plant growth stages. Alternatively, NGS studies could be conducted to evaluate the abundance of GTDs in nurseries so as to predict the future development of disease in established plantations. Finally, there is a need to build on the work performed by Bruez et al. (2017) [32] by using NGS technology to explore whether HWT has a long-term effect in the field.

In this study, we demonstrated that the short-term effects of HWT on GTD-related fungi are intricate and cultivar-dependent. Although our experiment revealed that HWT indeed causes changes in the GTD-related communities in plant tissues, these changes were not always characterized as a reduction in the abundance of GTD-related fungi. Instead, changes in fungal abundance were species-specific and mostly dependent on plant tissue type. Addressing industry concerns, we corroborate that HWT reduces plant biomass accumulation in the short-term; however, we were unable to disentangle the relationship between this reduction and the measured plant stress indicator, highlighting the need for further investigations. Overall, our study utilized modern sequencing technology to contribute to our understanding of the effects of HWT on the plant mycobiome. We also provided some clarification of the impact that HWT has on GTD-related communities and plant health, so that HWT can be better understood as a management option.

## Figures and Tables

**Figure 1 jof-08-00485-f001:**
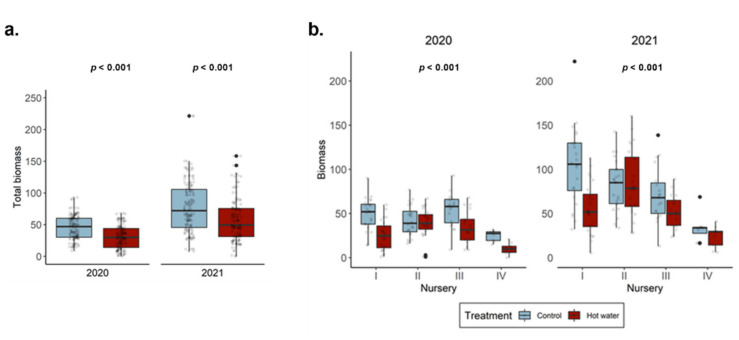
(**a**) Total accumulated biomass (g) of control and hot water-treated plants at planting (2020) and at harvest (2021) and (**b**) biomass analyzed by nursery of origin.

**Figure 2 jof-08-00485-f002:**
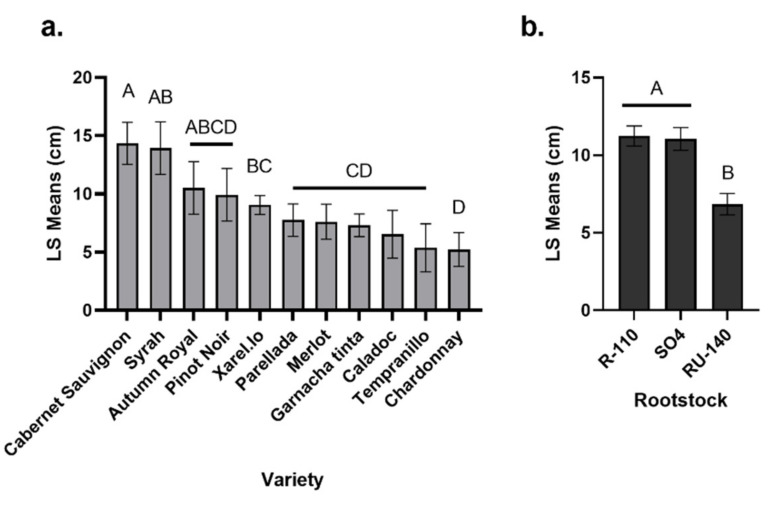
(**a**) Mean necrosis length (LS; cm) in graft unions for each variety; (**b**) Mean necrosis length in root collars of each rootstock. Post-hoc comparisons using a Student’s *t*-test were performed: values with different letters are significantly different.

**Figure 3 jof-08-00485-f003:**
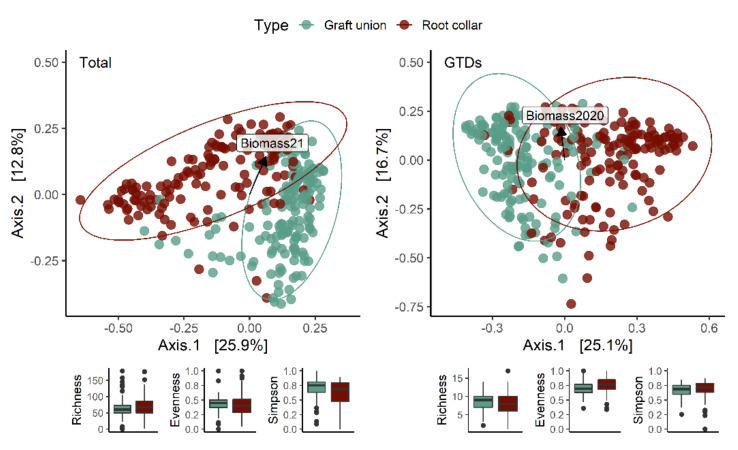
Principal coordinate analysis (PCoA) ordinations calculated with a Bray-distance matrix showing fungal community structures of all communities (**left**) and only GTD communities (**right**) by tissue type. Vectors indicate biomass accumulation for the whole plant in 2020 and 2021. The R^2^ and *p*-values shown are extracted from a PERMANOVA analysis performed with the adonis2 function in R, in which all design variables are included. Alpha diversity indices (species richness, evenness, and Simpson diversity index) are presented below each figure.

**Figure 4 jof-08-00485-f004:**
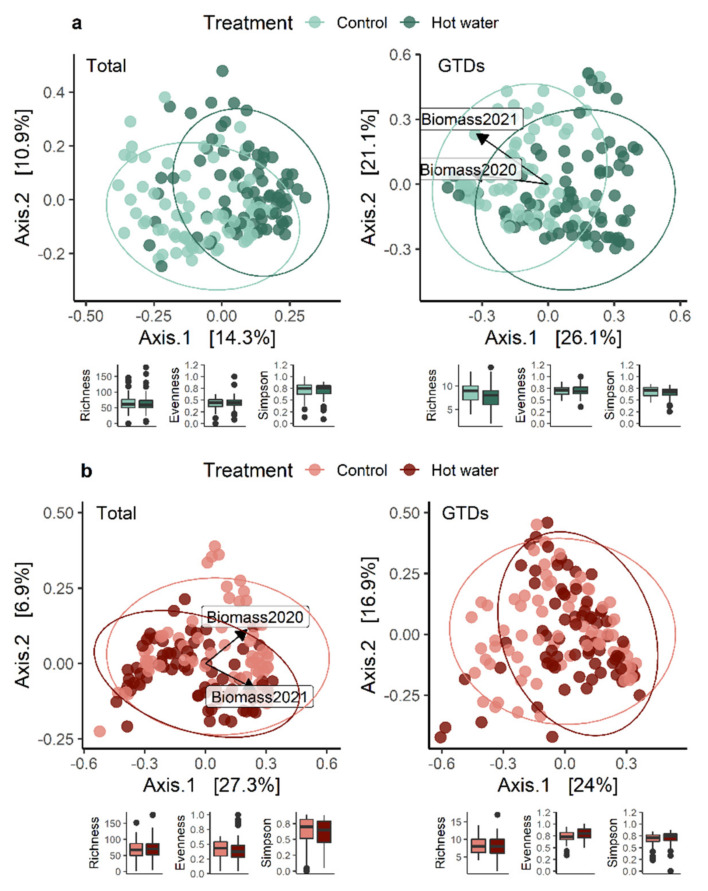
PCoA ordinations calculated with a Bray-distance matrix showing tissue-specific fungal community structures by treatment in (**a**) graft unions and (**b**) root collars. Vectors indicate biomass accumulation for the whole plant in 2020 and 2021. The R^2^ and *p*-values shown are extracted from a PERMANOVA analysis performed with the adonis2 function in R, in which all design variables are included. Alpha diversity indices (species richness, evenness, and Simpson diversity index) are presented below each figure.

**Figure 5 jof-08-00485-f005:**
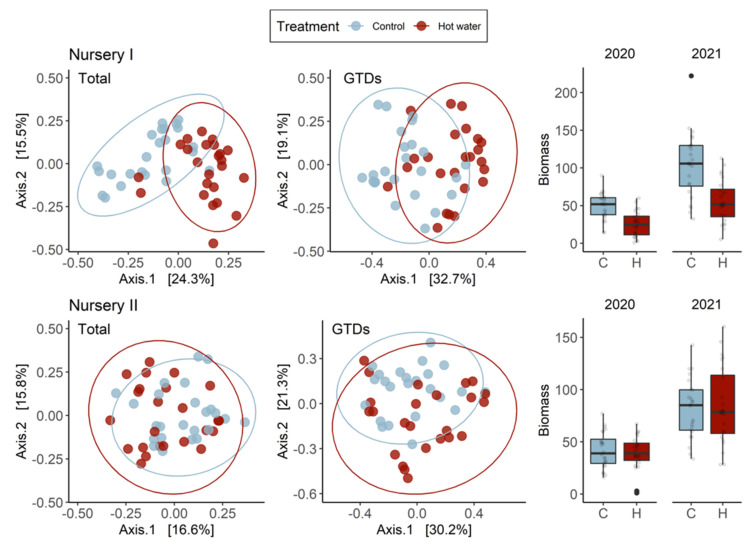
PCoA ordinations calculated with a Bray-distance matrix showing the contribution of nurseries I (**top**) and II (**bottom**) to total fungal community structures (**left**) and GTD community structures (**middle**) for each treatment in graft unions. The R^2^ and *p*-values shown are extracted from a PERMANOVA analysis performed with the adonis2 function in R. The biomass accumulation (g) of plants in 2020 and 2021 that originated from nurseries I and II (**right**) is shown on the right.

**Figure 6 jof-08-00485-f006:**
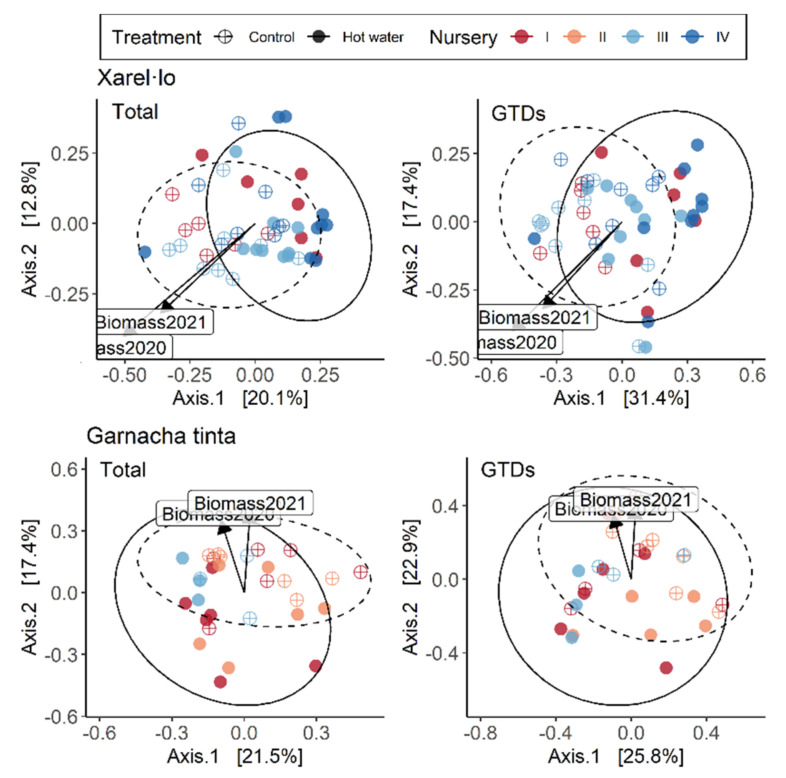
PCoA ordinations calculated with a Bray-distance matrix showing the contribution of varieties Xarel·lo (**top**) and Garnacha Tinta (**bottom**) to variation in the total fungal community (**left**) and the GTD-related fungal community (**right**) in plants originating from each nursery and for each treatment in graft unions. The R^2^ and *p*-values shown are extracted from a PERMANOVA analysis performed with the adonis2 function in R.

**Figure 7 jof-08-00485-f007:**
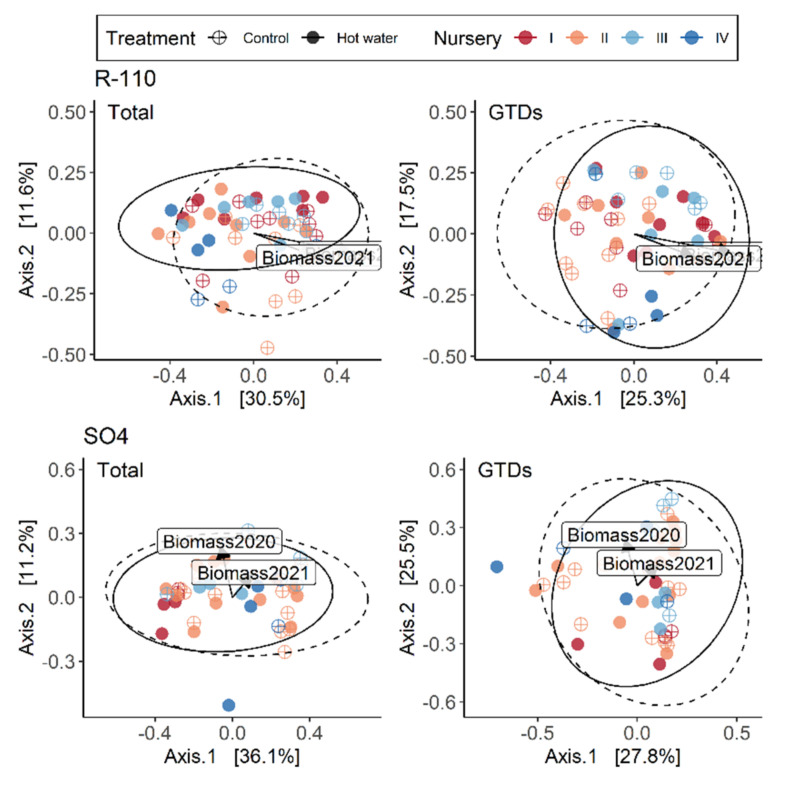
PCoA ordinations calculated with a Bray-distance matrix showing the contribution of rootstocks R-110 (**top**) and SO4 (**bottom**) to variation in the total fungal community (**left**) and the GTD-related (**right**) fungal community in each nursery and for each treatment in root collars. The R^2^ and *p*-values shown are extracted from a PERMANOVA analysis performed with the adonis2 function in R.

**Table 1 jof-08-00485-t001:** Study design showing the number of plants in each scion rootstock combination. Grafted plants were obtained from four different nurseries overall. At least two nurseries per color rootstock combination were included in replicates of six (i.e., three plants each of control and hot water treatment). Rootstocks shown are 110 Richter (R-110), 140 Ruggeri (RU-140), or Selection Oppenheim 4 (SO4).

	Rootstock	R-110	RU-140	SO4	Total
Red varieties	Autumn Royal	6			6
	Cabernet Sauvignon	6		6	12
	Caladoc		6		6
	Garnacha Tinta	12	18		30
	Merlot	6	6		12
	Pinot Noir			6	6
	Syrah			6	6
	Tempranillo	6			6
White varieties	Chardonnay	6	6		12
	Parellada	6	6	6	18
	Xarel·lo	18	12	18	48
	Total	60	54	48	

**Table 2 jof-08-00485-t002:** Contribution (r^2^) of nursery, variety/rootstock, and treatment to the variation in necrosis in graft unions and root collars. The annotation following the r^2^ values indicates the significance level: *** <0.001, ** <0.01, * <0.05, n.s. = not significant (>0.05).

Factor	Graft Unions (r^2^)	Root Collars (r^2^)
Nursery	0.051 *	0.036 n.s.
Variety	0.164 **	-
Rootstock	-	0.139 ***
Treatment	0.000 n.s.	0.004 n.s.
Full model	0.20 ***	0.18 ***

**Table 3 jof-08-00485-t003:** Contribution of tissue type, treatment, nursery, and variety/rootstock to total community variation (top) and GTD community variation (bottom) in graft unions, root collars, and both tissues combined (total). The annotation following the value indicates the significance level: *** <0.001, ** <0.01, * <0.05, n.s. = not significant (>0.05).

Total Community			
Factor	Graft Unions (r^2^)	Root Collars (r^2^)	Total (r^2^)
Tissue type	-	-	0.111 ***
Treatment	0.029 ***	0.028 ***	0.020 ***
Nursery	0.036 **	0.048 **	0.032 ***
Variety	0.084 ***	-	0.045 ***
Rootstock	-	0.022 *	0.007 n.s.
**GTD-Related Fungal Community**			
**Factor**	**Graft Unions (r^2^)**	**Root Collars (r^2^)**	**Total (r^2^)**
Tissue type	-	-	0.151 ***
Treatment	0.084 ***	0.015 *	0.029 ***
Nursery	0.036 **	0.057 ***	0.025 ***
Variety	0.089 **	-	0.048 ***
Rootstock	-	0.030 **	0.006 n.s.

**Table 4 jof-08-00485-t004:** Log change in the relative abundance (%) of GTD-related species in each tissue type between control and treated (HWT) plants. The significance of the difference between treatments is indicated for each tissue type (r^2^). The annotation following the r^2^ values indicates the significance level (*p*-value): *** <0.001, ** <0.01, * <0.05.

Graft Unions				
Species	Control (%)	HWT (%)	Log Change	r^2^
*Botryosphaeria* sp.	0.274	1.233	0.6532125	0.07
*Neofusicoccum parvum*	0.074	0.292	0.5961511	0.06
*Acremonium* sp.	1.657	3.652	0.3432083	0.31 ***
*Acremonium* sp. 2	0.019	0.039	0.312311	0.30 ***
*Phaeomoniella chlamydospora*	1.359	2.449	0.2557693	0.16 *
*Phoma* sp.	0.062	0.086	0.1421068	0.17 *
*Cylindrocarpon* sp. 2	0	0	0	0.07
*Nectria* sp.	0	0.048	0	0.08
*Pestalotiopsis* sp.	0	0.009	0	0.06
*Thelonectria* sp.	0	0	0	0.08
*Phaeoacremonium angustius*	0.71	0.578	−0.089331	0.13
*Cadophora* sp.	0.032	0.009	−0.550907	0.25 ***
*Cadophora luteo-olivacea*	5.143	1.166	−0.644518	0.26 ***
*Cylindrocarpon* sp.	0.057	0.01	−0.755875	0.07
*Diplodia* sp.	0.093	0.007	−1.123385	0.12
*Diaporthe* sp.	0.194	0.012	−1.20862	0.13
**Root Collars**				
*Thelonectria* sp.	0.001	0.045	1.6532125	0.06
*Acremonium* sp.	0.142	0.406	0.4562377	0.12 **
*Acremonium* sp. 2	0.002	0.004	0.30103	0.12 **
*Cylindrocarpon* sp. 2	0.014	0.021	0.1760913	0.02
*Botryosphaeria* sp.	0.018	0.024	0.1249387	0.06
*Phaeoacremonium angustius*	0.244	0.274	0.0503607	0.09 *
*Phoma* sp.	0.077	0.084	0.0377886	0.04
*Cadophora luteo-olivacea*	0.918	0.958	0.0185228	0.05
*Cadophora* sp.	0.006	0.006	0	0.05
*Nectria* sp.	0	0.003	0	0.03
*Pestalotiopsis* sp.	0.066	0	0	0.08 *
*Cylindrocarpon* sp.	0.851	0.678	−0.0987	0.03
*Diplodia* sp.	0.007	0.003	−0.367977	0.02
*Phaeomoniella chlamydospora*	1.095	0.177	−0.791441	0.18 ***
*Diaporthe* sp.	0.1	0.009	−1.045757	0.04
*Neofusicoccum parvum*	0.807	0.016	−1.702754	0.08 *

**Table 5 jof-08-00485-t005:** Contribution of each variety (top) to variation (r^2^) by nursery and treatment for total and GTD communities in graft unions and of each nursery (bottom) to variation (r^2^) by variety and treatment. The annotation following the r^2^ value indicates the significance level: *** <0.001, ** <0.01, * <0.05, n.s. = not significant (>0.05). Numbers in parentheses indicate the number of nurseries that supplied a particular variety (top) and the number of varieties supplied by each nursery (bottom).

Variety				
Variety	Total Community		GTD Community	
	Nursery (r^2^)	Treatment (r^2^)	Nursery (r^2^)	Treatment (r^2^)
Xarel·lo (3)	0.100 ***	0.051 ***	0.090 **	0.121 ***
Garnacha Tinta (3)	0.120 **	0.083 **	0.080 n.s.	0.080 *
Merlot (2)	0.167 *	0.119 n.s.	0.068 n.s.	0.182 n.s.
Chardonnay (2)	0.112 n.s.	0.166 *	0.079 n.s.	0.145 n.s.
Cabernet Sauvignon (1)	-	0.153 n.s.	-	0.120 n.s.
Tempranillo (1)	-	0.253 n.s.	-	0.383 n.s.
Caladoc (1)	-	0.197 n.s.	-	0.217 n.s.
Pinot Noir (1)	-	0.194 n.s.	-	0.303 n.s.
Syrah (1)	-	0.341 n.s.	-	0.422 n.s.
Autumn Royal (1)	-	0.216 n.s.	-	0.235 n.s.
Parellada (1)	-	0.128 n.s.	-	0.167 n.s.
**Nursery**				
**Nursery**	**Total Community**		**GTD Community**	
	**Variety (r^2^)**	**Treatment (r^2^)**	**Variety (r^2^)**	**Treatment (r^2^)**
I (6)	0.092 n.s.	0.140 ***	0.096 n.s.	0.146 ***
II (6)	0.159 **	0.060 ***	0.177 **	0.071 **
III (4)	0.113 *	0.086 ***	0.072 n.s.	0.101 ***
IV (1)	-	0.091 n.s.	-	0.108 n.s.

**Table 6 jof-08-00485-t006:** Contribution of each rootstock (top) to variation (r^2^) by nursery and treatment for total and GTD communities in root collars and of each nursery of origin (bottom) to variation (r^2^) by rootstock and treatment. The annotation following the r^2^ value indicates the significance level: *** <0.001, ** <0.01, * <0.05, n.s. = not significant (>0.05). Numbers in parentheses in indicate the number of nurseries that supplied a particular rootstock (top) and the number of rootstocks supplied by each nursery (bottom).

Rootstock				
Rootstock	Total Community		GTD Community	
	Nursery (r^2^)	Treatment (r^2^)	Nursery (r^2^)	Treatment (r^2^)
R-110 (4)	0.115 **	0.129 ***	0.050 **	0.048 **
SO4 (4)	0.182 ***	0.161 **	0.027 n.s.	0.011 n.s.
RU-140 (4)	0.091 n.s.	0.126 **	0.341 n.s.	0.260 n.s.
**Nursery**				
**Nursery**	**Total Community**		**GTD Community**	
	**Rootstock (r^2^)**	**Treatment (r^2^)**	**Rootstock (r^2^)**	**Treatment (r^2^)**
I (3)	0.140 ***	0.136 ***	0.031 n.s.	0.107 **
II (3)	0.054 *	0.058 n.s.	0.033 n.s.	0.083 *
III (3)	0.061 *	0.075 n.s.	0.072 **	0.095 *
IV (3)	0.065 n.s.	-	0.033 n.s.	-

**Table 7 jof-08-00485-t007:** Indicator species for Xarel·lo plants in both tissues for control and HWT plants. Correlation values (stat) and the statistical significance of the correlation (*p*-values) are shown. GTD-related indicator species are marked with a cross (†) and biocontrol species are marked with an asterisk (*). The annotation denoting the significance level is as follows: *** <0.001, ** <0.01, * <0.05.

Five Species Associated with Control Plants			
Indicator Species	Stat	*p* Value	Significance
*Acaromyces ingoldii*	0.559	0.0428	*
*Cadophora* sp. †	0.841	0.0001	***
*Cadophora luteo-olivacea_2* †	0.834	0.0001	***
*Ophiobolus* sp.	0.62	0.0128	*
*Pyrenochaeta* sp.	0.563	0.0339	*
**Eight species associated with HWT plants**			
*Acremonium* sp.*_13* †	0.786	0.0045	**
*Acremonium* sp.*_15* †	0.765	0.0225	*
*Cystobasidium lysinophilum*	0.733	0.032	*
*Fusarium* sp.*_3*	0.746	0.0195	*
*Fusarium* sp.*_6*	0.659	0.0071	**
*Fusarium* sp.*_22*	0.801	0.0333	*
*Stagonospora* sp.*_5*	0.754	0.0061	**
*Quambalaria cyanescens **	0.782	0.0448	*

## Data Availability

The data presented in this study are openly available in FigShare at https://doi.org/10.6084/m9.figshare.19714351 (accessed on 3 April 2022).

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
