# Peer review of "Hot Water Treatment Causes Lasting Alteration to the Grapevine (Vitis vinifera L.) Mycobiome and Reduces Pathogenic Species Causing Grapevine Trunk Diseases"

_jof, 2022, doi:10.3390/jof8050485_

Round 1

Reviewer 1 Report

The manuscript “Hot water treatment (HWT) causes lasting alteration to the grapevine (Vitis vinifera) mycobiome and reduces pathogenic species causing grapevine trunk diseases (GTDs)” is a very interesting contribution. However, it has been very difficult to carry out the revision since the document does not have lines. The references do not follow the MDPI format, and the English and its grammar should be improved.

I think the abstract should be improved, even more so if no conclusions were added. In this case, it would be very interesting to know the answers by variety and rootstock, which exist. In addition, the authors mention that this is a medium-term study and I think it is short-term, thereby, this should be edited in the text.

Abstract

More information about the treatments (control) and rotstocks should be added in the abstract section, accounting M&M, results and conclusions. In addition, if there were differences among rootstocks, this information should be added.

Introduction

First paragraph: The authors should replace "conspicuous" word by another

First paragraph: There are more problems associated to GTDs that should be added. In First paragraph: addition, the authors should add references along introduction section.

First paragraph: The reference format is not the correct. Please, edit it along the manuscript.

First paragraph: The first time that an acronym is mentioned, it should be described.

Third paragraph: Some references should be added in this sentence. Please, regard these following reports:

https://doi.org/10.1002/ps.6064

https://doi.org/10.3389/fmicb.2020.614620

Fourth paragraph: I think that it is important to add information about the killing temperature for pests and pathogens.

Fourth paragraph: Is that correct the term "fermentation" in this sentence?

Sixth paragraph: This paragraph is difficult to follow, please, remade it for a better understanding.

Material and methods

2.1 section: Rootstock names are incorrect, commonly are written as 110 Richter and 140 Ruggeri.

2.2 section: What’s "RT" means?

2.3 section: Basic climate information of the field trial should be added by each group of planting.

2.5 section: This sentence should be deleted. It is important only to focus on methodology.

2.5 section: More information about the measurement protocol should be added.

2.7 section: In 6 to 7 lines, this sentence should be edited for a better understanding.

Results

3.3 section: After Figure 4, this sentence should be edited for a better understanding.

3.3 section: Final paragraph, I think it could be better if the authors describe the differences by variety and rootstock.

3.4 section: Second paragraph, Varieties are proper names, so the correct form is to define as Garnacha Tinta.

Reviewer 2 Report

Dear authors,

manuscript "Hot water treatment (HWT) causes lasting alteration to the grapevine (Vitis vinifera) mycobiome and reduces pathogenic species causing grapevine trunk diseases (GTDs)" has practical importance in nursery production.

The introduction part should be reorganized to make the reading easier; there is a long list of previous experiments on some other popular solutions specific to GTD‐related fungi and biocontrol that should be shortened. In addition, the aims of the manuscript should be clearly indicated and highlighted in the introduction part.

Title: (Vitis vinifera L.)

Remove abbreviation from the title

Abstract and further through text (V. vinifera L.)

It is enough to put an abbreviation in one place in the text that could be an abstract

So please remove those in keywords and use just one kind through text

In the text, reference numbers should be placed in square brackets [ ], and placed before the punctuation; for example [1], [1–3] or [1,3].

Which four nurseries: be specific as there were probably two from one place „Four nurseries from the Catalan regions of either Girona, Tarragona or Barcelona participated in the study.“

What is RT water? „During this time, control plants were left to soak in RT water for 2 h.“

The discussion is too scattered and looks like results, without critical evaluation. I suggest the authors rewrite the discussion part.

Careful English language and style are needed “inciting the need for further investigation.”

Kind regards.

Round 2

Reviewer 2 Report

Dear Authors,

thank You for the new version of the manuscript.

Kind regards.